# Arboviruses: How Saliva Impacts the Journey from Vector to Host

**DOI:** 10.3390/ijms22179173

**Published:** 2021-08-25

**Authors:** Christine A. Schneider, Eric Calvo, Karin E. Peterson

**Affiliations:** 1Laboratory of Persistent Viral Diseases, Rocky Mountain Laboratories, National Institute of Allergy and Infectious Diseases, National Institutes of Health, Hamilton, MT 59840, USA; christine.schneiderlewis@nih.gov; 2Laboratory of Malaria and Vector Research, National Institute of Allergy and Infectious Diseases, National Institutes of Health, Rockville, MD 20852, USA; ecalvo@od.nih.gov

**Keywords:** viral infection, skin, immune enhancement, mosquito

## Abstract

Arthropod-borne viruses, referred to collectively as arboviruses, infect millions of people worldwide each year and have the potential to cause severe disease. They are predominately transmitted to humans through blood-feeding behavior of three main groups of biting arthropods: ticks, mosquitoes, and sandflies. The pathogens harbored by these blood-feeding arthropods (BFA) are transferred to animal hosts through deposition of virus-rich saliva into the skin. Sometimes these infections become systemic and can lead to neuro-invasion and life-threatening viral encephalitis. Factors intrinsic to the arboviral vectors can greatly influence the pathogenicity and virulence of infections, with mounting evidence that BFA saliva and salivary proteins can shift the trajectory of viral infection in the host. This review provides an overview of arbovirus infection and ways in which vectors influence viral pathogenesis. In particular, we focus on how saliva and salivary gland extracts from the three dominant arbovirus vectors impact the trajectory of the cellular immune response to arbovirus infection in the skin.

## 1. Introduction

Arbovirus infection involves complex interactions between the virus, the vector, and the host. In extracting a blood meal, the vector induces skin damage at the bite site, initiating a tissue repair response from the host. Simultaneous deposition of an arbovirus into the bite site sets off a battle of viral immune evasion and host immune defense. Blood-feeding arthropods (BFAs) further complicate these interactions because their saliva contains highly bioactive salivary proteins that can block, enhance, or alter each of these interactions. This review will examine these relationships (virus, vector, and host) with a focus on BFA saliva and salivary proteins.

## 2. Arboviruses

Arthropod-borne viruses (arboviruses) can be found in multiple virus families, although the majority cluster within three: Bunyaviridae, Flaviviridae, and Togaviridae (Figure 1). Arboviruses within the Bunyaviridae family have extensive genus and vector diversity, with viruses from this family transmitted to humans by sandflies, mosquitoes, and ticks [1]. The severity of bunyavirus-induced disease in humans also fluctuates from asymptomatic to severe disease, including seizures, coma, and in rare cases, death. The Flaviviridae family contains one genus of arboviruses, all of which are transmitted to human hosts by mosquitoes or ticks [2]. As a genus, flaviviruses are the most prevalent and widely studied arboviruses, with extensive recent focus placed on Dengue virus (DENV) and Zika Virus (ZIKV). ZIKV is capable of vertical and sexual transmission in addition to being vector-borne, although these are not the dominant transmission mechanisms [3]. Flaviviruses are globally distributed and can cause widespread morbidity and mortality. Togaviridae contains a single genus, alphaviruses, which are exclusively mosquito-borne, with *Aedes* and *Culex* mosquitoes serving as important vectors [4,5]. These alphaviruses have less world-wide distribution and cause fewer cases of human disease overall than flaviviruses [3,6]. However, a recent increase in cases of severe febrile illness has been observed with the alphavirus, Chikungunya virus (CHIKV), as mosquito vector populations have shifted due to changing climates and infection of different mosquito species [7,8]. Additional arboviruses have been identified outside of these families and include both tick- and sandfly-borne viruses (Figure 1). Overwhelmingly, individual arboviruses are transmitted by a single type of vector (i.e., mosquitoes, ticks, or sandflies), with the exception of Vesicular Stomatitis virus (VSV; Rhabdoviridae), which is carried and transmitted readily by both sandflies and mosquitoes [9,10]. It is unclear why most viruses are not carried by multiple vectors simultaneously, but this may be related to host biology or a function of geographical location. Indeed, some viruses are vector-restricted even within an arthropod family. La Crosse virus (LACV; Bunyaviridae), for example, can infect both *Aedes triseriatus* and *Aedes hendersoni* mosquitoes and is passed transovarially to mosquito offspring by both, but is only readily saliva-transmitted to hosts by *A. triseriatus* [11]. Despite being part of diverse virus families, arboviruses share many functional similarities in how they interact with the mammalian host and arthropod vector.

### Arbovirus Infection of Vectors

Successful transfer of arboviruses to mammalian hosts begins first with infection of the arthropod vector. After taking a bloodmeal from an infected host, the virus replicates within the arthropod midgut and eventually reaches the salivary glands where it is transferred to the host during feeding. Although most research focus is placed on host pathology during mammalian infection, arboviruses can also impact the vector in both beneficial and harmful ways. Flaviviruses like DENV and WNV (Flaviviridae) reach the mosquito central nervous system (CNS), resulting in altered feeding behaviors that negatively impact successful bloodmeal acquisition [12,13,14,15]. Rift Valley Fever virus (RVFV; Bunyaviridae) similarly alters feeding behavior, but also reduces egg output and is ultimately lethal to mosquitoes [16]. Conversely, Sindbis virus (SINV; Togaviridae) is neuroinvasive within the mosquito, resulting in mosquitoes gaining resistance to the insect repellent compound DEET, reducing options available for preventing mosquito bites in humans [17,18]. Not all mosquito arboviruses alter feeding behaviors, however. Feeding disruption is only transiently observed with La Crosse virus (LACV; Bunyaviridae), a mosquito-borne virus in the same family as RVFV [16,19]. The virus’ ability to induce pathology in the vector species may be a unique feature to mosquitoes. There are no currently reported benefits or detriments to ticks or sandflies from virus infection, although many of these questions are just beginning to be explored.

## 3. Arbovirus Infection of the Host

During feeding, BFAs inject protein-rich saliva as an aid in blood extraction, carrying arboviruses that are deposited into host skin. Vector saliva intercepts important coagulation pathways in the host but also can enhance viral dissemination and may increase pathogenesis of virus infections [20,21]. For example, neurological disease is more severe in experimental mouse models of Semliki Forest virus (SFV; Togaviridae) and West Nile virus (WNV; Flaviviridae) infection when salivary gland extract (SGE) is co-injected with the virus [22,23]. Dengue virus (DENV; Flaviviridae) replicates to higher viral titers in the skin in mice injected with *A. aegypti* SGE compared to DENV infection alone [24]. Similarly, Zika virus (ZIKV; Flaviviridae) titers are higher in many peripheral organs and the brain when inoculated into mice alongside isolated *A. aegypti* salivary protein LTRIN [25]. In some cases, salivary enhancement occurs even if virus and saliva are not injected at the same time. For WNV, mosquito saliva enhances viral titers even if injected 12 h after the virus, so long as it is spatially close to the initial virus injection site [26]. Interestingly, there appears to be a vector species-specific enhancement in some cases, with higher viral titers and more rapid death when RVFV (Bunyaviridae) is administered with *A. aegypti* SGE but not *C. pipens* SGE despite both being competent vectors for transmission [27,28,29,30]. Notably, salivary enhancement is also observed with tick-borne Powassan virus (POWV; Flaviviridae) and tick-borne encephalitis virus (TBEV; Flaviviridae), revealing that this is not a unique feature reserved solely to mosquitoes [31,32].

In addition to heightened viral pathology and titer, SGE has the potential to increase viral spread to new hosts. Naïve ticks fed on guinea pigs infected with Thogoto virus (THOGV; Orthomyxoviridae) become virus-positive more often when the host guinea-pigs are injected with tick SGE than virus injection alone [33]. Naïve ticks can become infected with THOGV after feeding on a naïve mouse if they co-feed alongside infected ticks—even in the absence of host seroconversion [33]. This spread in the absence of viremia in the host is of particular interest as ticks are frequently found in tightly-spaced clusters when feeding on natural virus reservoir species, suggesting localized transmission between ticks [34]. This is not a vector-specific observation as it also occurs in mouse models of WNV. If naïve mosquitoes feed on mice within 40 mm of an infected mosquito feeding site and no more than 45 min after the infected mosquito, naïve mosquitoes can become infected with virus [35,36].

Many theories have been put forth to explain why saliva and SGE increase local viral titer, including enhancing viral fusion to certain cell types, as has been shown with DENV in vitro [37]. Tick salivary proteins also prevent upregulation of interferon (IFN) β by infected dendritic cells (DCs), which leads to unchecked TBEV replication in vitro [38]. Evidence suggests viral dissemination in mammalian hosts may also be enhanced through immune modulation, positively influencing the mobilization and motility of immune cells from the skin to the lymph node (LN) and promoting viral spread [24,37]. Increased motility of infected immune cells may explain the spread of THOGV between co-feeding ticks despite a lack of detectable host viremia [33].

### 3.1. Feeding Differences of Vectors

Hematophagous arthropods rely on acquisition of blood meals as a rich source of proteins and heme for producing eggs and, in ticks, for progressing to the next stage of development [39]. In general, biting arthropods obtain blood from hosts by the same general mechanism, making use of sharp mouthparts to penetrate the host skin to reach blood vessels. During this process, protein-rich saliva is injected into the skin, facilitating blood extraction by intercepting the normal processes of hemostasis that prevent blood loss. Mosquitoes, ticks, and sandflies have therefore evolved mechanisms to prevent coagulation, limit vasoconstriction, and suppress pain receptors to facilitate feeding off mammalian hosts. The specific mechanisms used are, in part, tailored by the unique feeding style employed by each BFA.

#### 3.1.1. Mosquitoes

The three largest mosquito genera, *Culex*, *Aedes*, and *Anopheles*, all serve as competent vectors for viral diseases. Mosquitoes have substantial genetic diversity between genera as well as the viruses they carry. *Aedes* mosquitoes are the dominant vector for mosquito-borne diseases, although *Culex* and *Anopheles* also carry pathogens that cause disease in humans [40,41]. Divergence within the mosquito genera has led to strains with substantially different feeding timetables and behaviors; however, all mosquitoes feed using the same physical mechanisms. Two thin serrated cutting projections called maxillae are used to penetrate through the thinner outer epidermis into the dermis layer below [42]. Sandwiched between the maxillae is a structure of two attached tubes, one for extracting blood and one which passes saliva into the host (Figure 2). The labrum resembles a needle with a beveled tip that has sensors for detecting blood vessels and ultimately for extracting blood once a vessel is located [42,43,44]. Laying on top of the labrum is the hypopharynx through which saliva is continually secreted [43]. The process of locating a vessel sometimes requires multiple penetrations into the skin for successful feeding to begin, but the penetration of such a small microneedle structure often elicits little pain [45]. Once a vessel is located and cannulated, most mosquitoes feed to repletion within 1–2 min [46]. Both male and female mosquitoes feed on plant juices for most of their nutrients, but female mosquitoes feed on mammals and birds when ready to produce eggs. The composition of the sialome (i.e., salivary proteome) changes to match this shift in food source [47]. Proteomic comparisons between sugar and blood-feeding female *Aedes aegypti* mosquitoes revealed increased expression of proteins intercepting host responses to biting, including vasoconstriction, pain sensation, and cytokine production, when females were fed on blood [47]. This represents the intensely dynamic nature of vector saliva, including the ability to tailor the composition to match shifting needs so that essential proteins are only expressed when most needed.

#### 3.1.2. Ticks

Disease-transmitting ticks are subdivided into two main families based on their body composition—hard-bodied (Ixodidae) and soft-bodied (Argasidae). Members of both families are obligate blood-feeders, although Ixodidae ticks actively search out hosts for extended feedings (days to weeks) while Argasidae reside within host nests, allowing for repeated brief feedings (hours) from developing rodents and birds. As a result, Ixodidae have increased potential to transmit bacterial pathogens to hosts as pathogen transfer increases with longer duration of skin attachment [48]. Long duration feeding is not required for transfer of POWV, which reach hosts within the first 15 min of attachment to mouse skin [49]. Both sexes of ticks ingest blood to progress between developmental stages, after first emerging from eggs through nymph stages to adults [50]. Adult females require a blood meal as a rich source of heme for producing viable eggs, and many adult males require a blood meal to produce viable sperm [39,51]. In at least two tick species, males and females have different salivary protein profiles [52,53,54], although it has not been investigated if this impacts viral transmission.

Both hard and soft-bodied ticks have large cutting mouth parts called chelicerae to saw through the epidermis and deep into the dermis to access blood [55]. The back and forth sawing motion of the chelicerae forms a wound and results in the formation of a blood pool into which the barbed single-channel hypostome feeding tube is inserted [55] (Figure 2). Histology of mouse skin performed during active tick feeding revealed that the hypostome can even reach the skin fat cells on the border of the hypodermis [56]. Remarkably, ticks are able to attain the same depth of penetration across the nymph and adult stages, despite having vastly different body sizes [57]. However, the wound size and the amount of blood pooling into it is proportional to the size of the tick as well as the tick developmental stage, with adult ticks able to extract more whole blood than smaller nymphs [57]. Although nymphs have a smaller feeding volume, they are still able to acquire viral and bacterial infections from infected hosts so long as the feeding duration is sufficiently long for a given pathogen [48].

Often, a single skin penetration is sufficient to establish feeding for the duration of the attachment to the host—a period of a few hours (soft-bodied ticks) to weeks (hard-bodied ticks), depending on the species [58]. To enhance long-term attachment to the host, some species secrete a thick cement-like substance to surround the entire surface between their body and the host to prevent detachment [59]. This firm attachment facilitates tick engorgement, where female hard-bodied ticks may swell 30–100x their pre-attachment body weight in a feeding session [50]. During pauses in blood extraction, saliva is secreted back into the host, with the longer feeding hard-bodied ticks returning up to 74% of the water extracted from the total blood meal volume [60]. This saliva exchange uses the salivary glands to concentrate the removal of the most essential blood components, while also providing a mechanism for direct host manipulation through injection of salivary proteins that aid in feeding [60]. Tick-derived components in the saliva also help maintain the fluidity and integrity of the blood pool by interrupting clotting and vasoconstriction while also preventing immune cell repair to the wound area and suppressing pain [61]. As with mosquitoes, tick saliva is highly dynamic, but a shift in the sialome content occurs not because the food source changes, but instead by the duration of host attachment. In addition, compared to mosquitoes who feed only briefly, tick saliva is more complex, containing many more secreted proteins of uncharacterized function [62]. This is suggested to result from the need to interfere with numerous extensive processes involved with being long-term feeders, including interrupting tissue repair programs designed to close the wound [62]. Simo et al. have compiled a nice in-depth review of vasoactive and functionally active proteins identified to date in tick saliva [61].

#### 3.1.3. Sandflies

Around 1000 species of sandflies have been identified with the arbovirus-relevant species found within two genera: *Phlebotomus* and *Lutzomyia* [63]. Although more commonly vectors for parasitic diseases like Leishmaniasis, infected female sandflies can transmit phleboviruses during blood feeding, and cases are often found clustered geographically during an outbreak [64]. The sandfly proboscis is structurally and functionally similar to the mosquito, with a two-channeled extraction—secretion mechanism surrounded by sharp maxillae [65]. However, the feeding style employed is much more similar to ticks. Female sandflies use the extensive sharp barbs on their maxillae and mandibles to cut a wound in the skin, creating a blood pool at the very surface of the dermis [65,66]. This wound formation is essential given that the labrum is too short to extend deeply into the dermis. This results in a comparatively shallow blood pool formed primarily in the epidermis, although saliva penetrates down into the dermis, inducing vascular leakage to supply the blood pool (Figure 2) [67]. In contrast to ticks, however, the blood meal is rapidly extracted over a period of minutes in a timescale similar to mosquitoes. This short duration makes maintenance of the blood pool less critical, although efficient interruption of hemostasis is still continuous and essential. Hematophagous female sandflies have saliva with 20x the protein concentration of non-blood feeding males, and work is ongoing to characterize the sialome of different sandfly species [68]. The very small body size of sandflies (<3 mm) reduces the likelihood of host detection during feeding, which is an important quality as the laceration formed by the bite does cause sharp pain in the host which is not inhibited by sandfly saliva [69].

### 3.2. Secretion of Saliva by Vectors

Despite differences in penetration depth and style of feeding, the injection of enzyme-rich, bioactive saliva into host skin is an essential component of arthropod blood feeding. For all three vectors, saliva injection begins from the moment of skin penetration and continues until the blood meal is finished [21,61,67,70,71]. Importantly, saliva is deposited differentially based on biting style—mosquitoes inject saliva both into the epidermis and dermis during initial skin penetration as well as adjacent to the blood stream once a vessel is successfully cannulated as feeding begins (Figure 2) [70,71]. Sandflies deposit saliva primarily into the epidermis due to their short mouthparts, but they are able to seep deeper into the epidermis and dermis as hyaluronidase enzymes and salivary proteins liquify the surrounding environment (Figure 2) [72,73,74,75,76,77]. Ticks, by sawing directly into the dermis through the epidermis, secrete saliva through both dermal layers and continue saliva injection for the entire long duration of feeding, resulting in substantial saliva coverage of the bite area (Figure 2) [21,71]. This contrast in saliva injection location has the potential to differentially influence the local immune response, given the carefully arranged cellular immune organization in different skin layers.

## 4. Vector Transmission Impacts on the Anti-Virus Host Response

### 4.1. The Skin Immune Environment

The skin serves as the first line of defense against insect-borne viruses, with specialized cells primed to react to both the cellular damage caused by the arthropod bite and to any infectious organism the bite delivers. The skin is divided into three structural layers—the thin epidermis is at the top, bridging the distance between the air interface and blood-vessel rich dermis below [78]. In the absence of infection, the epidermis contains only keratinocytes and Langerhans cells (LCs), both of which perform surveillance functions reminiscent of myeloid cells [78]. The highly vascularized dermis contains a much wider repertoire of immune cells including dermal macrophages, T and B cells, dendritic cells (DCs), and additional LCs [79,80]. The hypodermis is populated almost exclusively with adipose cells, although there are resident immune cells at low levels in the absence of inflammation [81].

During infection or injury, epidermal cells initiate the first reaction to foreign material through detection by pattern-recognition receptors (PRRs) expressed on host cells. These initiate cytokine production and the release of anti-microbial peptides [82,83]. Dermal innate and adaptive immune cells, along with DCs and resident macrophages, amplify inflammation during infection or wounding [80,84,85]. Chemokines recruit neutrophils, monocytes, and other peripheral immune cells to infiltrate into the highly vascularized dermis, followed by migration into the epidermis or hypodermis as necessary [83,84,85]. These processes work to reduce viral load and facilitate clearance of the infection. Epidermal and dermal LCs perform hybrid functions often ascribed to both macrophages and DCs, including migrating to the LN to present antigens to initiate adaptive immunity [83,86]. As the innate immune phase wanes and gives way to the adaptive response, T and B cells work to target virus-infected cells and produce antibodies specifically developed against the virus [79,80]. These processes all work together to resolve the infection and promote healing of the bite site.

### 4.2. Arboviruses Hijack Skin Immune Mechanisms

Arboviruses have evolved means of interrupting the development of a normal skin immune response by directly infecting and hijacking skin immune cells. Focusing first on the epidermis, Keratinocytes are permissive to infection with mosquito-borne flaviviruses, including DENV and WNV, both in vitro and in vivo, potentiating them as a replicative niche in the skin [87,88]. In addition, epidermal Langerhans cells are readily infected by mosquito-borne DENV, WNV, and CHIKV as well as tick-borne TBEV [20,89,90]. Because LCs are migratory, moving from the epidermis through the dermis and even to the draining LN, they may not only serve to amplify viral load, but also as a transport mechanism for the virus to the rest of the body. Indeed, infected migratory LCs are thought to contribute directly to the transfer of TBEV from infected ticks to naïve ones feeding on the same naive host through their migration [32].

Ticks and mosquitoes, by virtue of their longer mouthparts, are capable of also depositing virions below the epidermis and deep into the dermis. Here, virus particles encounter a more diverse repertoire of immune cells, increasing the diversity of cells available for viral invasion. Tick-borne POWV is deposited deeply into the skin by infected ticks, and the virus is detectable, flanking the bite site in the dermis and hypodermis, co-localized with myeloid and T cell markers, suggesting these cells are actively infected [91].

Depending on the cells infected, virus deposited in the dermis is not restricted to remain there. DENV, Crimean—Congo Hemorrhagic Fever virus (CCHFV), and TBEV replicate readily within dermal DCs, which are capable of migration, potentiating their transport to the descending LN [90,92]. As the infection progresses, immune cells become activated and cytokine production recruits additional hematopoietic-origin monocytes from the blood. Monocytes are permissive to TBEV infection and are shown to shuttle virus to peripheral tissues and even into the CNS [93,94,95]. ZIKV preferentially targets monocytes in both adult and fetal infection, modulating adhesion molecule expression that may assist with viral dissemination [96,97,98,99]. Similarly, DENV targets monocytes for infection and may participate in inducing endothelial damage, precipitating vascular permeability [100]. Thus, arboviruses take advantage of the dermal immune mechanisms essential for anti-viral defense to facilitate their access to the rest of the host.

### 4.3. Salivary Gland Protein Suppression of Skin Host Immune Responses

Intrinsic viral mechanisms can influence immune cell activation and function in the skin, but with vector-borne viruses, this is only one part of the overall infection picture. In addition to alterations in viral load and dissemination, vector saliva and SGE also exhibit immunomodulatory properties that can influence the trajectory and pathogenesis of viral infections [101]. This is particularly true early in infection where the innate immune system is the primary response to viral infection [102]. This is accomplished in three main ways: (1) modulation of cellular activation at the infection site in the skin, leading to (2) alterations of cytokine production, both related to propagation of inflammatory signaling or cellular recruitment, and (3) manipulation of immune cell motility, both to the skin and away to the skin-draining LN. These three processes are intertwined, and disentangling which event occurs first is often challenging. Despite profound evolutionary divergence between ticks, sandflies, and mosquitoes, their saliva retains remarkable functional conservation in how it interacts with the mammalian immune system, discussed below. While a nice body of literature has been generated characterizing SGE with mosquito and tick viruses, there is a relative lack of studies on sandfly saliva during viral infection. Sandflies, while the primary vectors for several phleboviruses, are much more commonly studied for their dissemination of *Leishmania sp.*, an intracellular parasite and cause of leishmaniasis worldwide. Although parasitic infections are largely beyond the scope of this review, there are potentially important parallels to be drawn by discussing a few aspects of sandfly saliva during *Leishmania* infection.

#### 4.3.1. Modulation of Innate Immune Activation

Across the three arthropod families, SGE widely suppress the initial post-bite innate immune response. This is accomplished at multiple levels, by dampening activation of skin-resident macrophages and DCs and by suppressing immune cell recruitment to the skin. Both maturation and activation of DCs by standard immune stimulants, such as poly I:C, CpG, or LPS, in the absence of infection are suppressed by tick SGE [103,104]. At least a portion of this suppressive function is due to salivary Prostaglandin E2, including suppressed cytokine production and reduced ability to stimulate T cells [105]. *Aedes aegypti* saliva also induces diminished DC—T cell stimulation but is the result of murine T-cell apoptosis in the absence of any alteration of DC maturation [106]. Macrophages in culture exposed to sandfly saliva have a blunted response to IFN-γ, producing less nitric oxide (NO), and leading to reduced ability to kill *L. major* parasites [107]. Tick SGE also lowers the capacity for re-activation with antigen when DCs are exposed to saliva during maturation, making them deficient at responding to additional exposures [103]. Finally, macrophages cultured with either WNV or SINV and treated with SGE from *A. aegypti* produced less IFN-β and iNOS compared to untreated infected cells [108], demonstrating that vector salivary components are capable of widespread immune suppression.

#### 4.3.2. Alterations of Cytokine Production

SGE-mediated suppression of myeloid cell activation might help to explain how salivary proteins induce differential cytokine profiles during virus infection. Tick saliva reduces pro-inflammatory cytokine production in mice both if given alongside *Borrelia* spirochetes and if used to prime in advance of infection [109]. Mosquito SGE reduced viral spread in mice infected with Semliki Forest virus (SFV) by suppressing neutrophil-produced CCL2 responsible for recruiting pro-inflammatory monocytes [22]. One potent inhibitor of cytokines in mosquitoes is the salivary protein LTRIN, which acts by blocking of downstream activation of the lymphotoxin B receptor [25]. This inhibition prevents cytokine production downstream of NFκB activation, resulting in reduced neutrophil and macrophage recruitment to the blood during mouse models of ZIKV infection [25]. The reduction in chemokine expression also corresponds with decreased cellular recruitment to mouse skin and LN observed during viral infection in the presence of SGE [25].

Endothelial cells are sensitive to many pathogens and inflammatory environments [110]. Their activation during the early immune response can induce substantial cytokine production essential for facilitating immune cell recruitment and infiltration to sites of inflammation [110,111]. Isolated tick salivary protein Longistatin binds to an endothelial cell receptor used to detect foreign glycans, effectively preventing activation and cytokine produced by endothelial cells in the skin [112]. In contrast, however, sandfly salivary proteins LuloHya and Lundep enhance chemokine production from endothelial cells, contributing to vascular leakage and immune cell recruitment [73].

#### 4.3.3. Prevention of Complement Activation

Activation of the complement cascade contributes to mobilization of many arms of the innate immune response, including increasing cytokine production, enhancing chemotaxis, and destroying viruses and infected cells for opsonization [113]. In arboviral infections, complement can be either protective against virus infection or contribute to viral pathology. Complement activation helps curb the spread of SINV and WNV by assembly of the membrane attack complex (MAC) that initiates lysis of viral particles or virus-infected cells [113,114,115,116]. Complement activation also results in enhanced antibody production in WNV and YFV, resulting in viral clearance [113,116,117,118]. However, in DENV infection, excessive cleavage of early complement proteins contributes to the development of vascular leakage and pathology in both animal models and patient samples [119,120,121,122].

BFA saliva contains proteins that can directly antagonize complement activation. At least three separate sandfly proteins have been identified that interfere with deposition of complement factors in both the alternative and classical pathways [123,124,125]. Similarly, *Anopheles sp.* mosquito saliva contains albicin, a protein directly antagonizing C3 cleavage and preventing full activation of the alternative pathway [53,126]. Perhaps most extensively studied, tick saliva contains nearly a dozen identified proteins that inhibit the classical or alternative complement pathways or block initial recognition by the lectin pathway [127]. Prevention of complement activation is thought to aid in BFA feeding by preventing inflammation and anaphylaxis that would be otherwise induced at the bite site. Whether these factors in BFA saliva enhance arbovirus transmission or augment host pathology is still under investigation and may vary depending on both host and vector.

#### 4.3.4. Inhibition of Immune Cell Recruitment

Saliva and SGE may also be able to directly inhibit macrophage and monocyte motility. Rodriguez et al. noted decreased migration of human monocyte-derived DCs during recruitment by CCL19 when *H. marginatum* tick SGE was added to the cultures [92]. Similarly, mouse peritoneal exudate cells are prevented from migrating in transwell plates when cells are exposed to isolated tick salivary protein Longistatin [112]. RNAi-mediated blockade of Longistatin in ticks just before feeding on mice results in much higher immune cell recruitment to the bite site than in non-treated ticks [112]. Loss of Longistatin in tick saliva also reduces blood pool formation in mouse skin and prevents successful feeding, further emphasizing the importance of proper immune suppression to allow effective vector blood extraction [112]. Isolated tick salivary proteins directly inhibit recruitment of neutrophils during mouse skin infection while also dampening their effective use of reactive oxygen [128,129]. Tick saliva also contains a homologous protein to mammalian macrophage inhibitory factor (MIF), which blocks human macrophages and monocytes from being recruited to an inflammatory stimulus [130,131,132,133]. This is similar to reduction in mouse footpad swelling by injection of tick-derived Amregulin [134]. This direct motility inhibition has not been extensively discussed in the literature for mosquito or sandfly saliva but may represent a unique feature of tick feeding.

### 4.4. Sandfly SGE Enhances Chemotaxis to the Bite Site

In contrast to the above studies showing inhibition of innate immune responses, SGE from sandflies may instead promote activation of certain components of the immune response. For example, SGE from sandflies is chemotactic to mouse monocytes, influencing their motility towards the site of saliva injection through induced local expression of monocyte chemokine CCL2 [135,136]. This was suggested to increase the potential spread of the human parasite *Leishmania*, as the parasite preferentially replicates within macrophages, which can differentiate from infiltrating monocytes [135,136]. Indeed, injection of salivary gland lysates along with parasites results in mouse skin lesions that are larger and have higher parasite burdens than parasite injection alone [137]. Saliva exposure also induces more monocytes and NK cells to remain at the bite site up to a week post infection [138]. This represents an example of saliva-induced enhancement of host immunity not observed in the other two vector groups but suggests that sandfly evolution may be influenced in part by the needs of the predominant infection they carry. Additionally, mice are protected from infection by *L. major* by being bitten by uninfected *P. duboscqi* sandflies [138]. This protection is due to saliva exposure creating a primed environment for NK and T cell recruitment, including substantially greater activation and cytokine production [138]. Much of this function appears to be due to the protein Maxadilan, shown to be an effective vaccination strategy against *L. major* in vivo [139]. These results provide an interesting contrast to tick and mosquito phenotypes, but it remains to be established if these hold during viral infection in sandflies. Currently, there are no studies of sandfly sialomes during phlebovirus infection, but that work would provide an interesting potential for comparison with other viral vectors.

### 4.5. Modulations of Adaptive Immunity

Arthropod saliva can also affect the adaptive immune response. Tick saliva reduces T cell proliferation when co-incubated with DCs, thus preventing normal T cell mediated immune responses from developing [109]. This was mirrored by treatment with SGE from *A. aegypti*, with reduced mouse T cell proliferation and cytokine production observed with Concanavalin A activation [140]. This effect is mosquito-species-specific and not observed with *Culex* sp. SGE, which is toxic to murine T cells [140]. T cell recruitment is suppressed when mosquitoes are allowed to feed on mice near sites of WNV injection in the skin, reducing the effectiveness of skin-local adaptive immunity [108]. Finally, SGE from sandflies, mosquitoes, and ticks skews the development of naïve T cells from a normal anti-viral Th1 response to Th2 both in vitro and in vivo, reducing their effectiveness in targeting pathogens [141,142]. Mosquito SGE alters cytokine production (including IL-4, IL-10, and IFN-g) from re-stimulated T cells ex vivo [143]. In vivo, both tick and sandfly salivary gland lysate injected into the skin upregulate IL-4 mRNA in the skin-draining LN [141]. Furthermore, tick cystatin proteins Iristatin and Sialostatin suppress normal cytokine production from T cells, resulting in reduced cellular recruitment and inflammation in mice in vivo [144,145]. Iristatin also prevents T cell activation during antigen recognition and blocks efficient T cell recruitment in mice [144]. This T cell phenomenon is not only observed in rodent models, as human cells also experience suppressed Th1 cytokines in favor of Th2 when exposed to sandfly SGE and isolated sandfly salivary protein Maxadilan [146]. The interruption of normal adaptive immunity by SGE and BFA saliva has the potential to widely influence arbovirus pathogenesis and resolution of infection.

## 5. Comparing Vectors—Overall Lessons to Be Learned and Looking Ahead

The three vectors discussed in this review employ different feeding styles to extract blood, harbor different viruses across different virus families, and are only distantly related. Despite this, there is profound functional conservation in anti-immune mechanisms employed by these vectors through their saliva. All three vectors withdraw blood after penetrating the skin but do so with varying degrees of tissue damage and from different places in the skin. Because the skin contains functionally distinct layers, this placement difference means biologically active salivary molecules and arboviruses will contact different resident and immune cells post deposition. The upper epidermis is full of innate immune sentinels poised to detect damage and pathogens while the lower dermis contains semi-resident populations of both innate and adaptive immune cells. Ticks, mosquitoes, and sandflies all inhibit T cell proliferation and production of key inflammatory cytokines using their saliva and salivary protein repertoire. It is essential for ticks to interrupt immune cell recruitment and tissue repair which would prevent their long-term feeding; however, short-duration feeding mosquitos and sandflies also alter cell chemotaxis and activation with their saliva. These immune-modulation functions result in similar viral outcomes—an overall enhancement of arboviral disease. In some cases, as with mosquito saliva and dengue virus, this occurs because of a direct interaction between salivary proteins and virus particles, resulting in improved cell entry and replication [24]. In other instances, it is indirect, resulting instead because of poor immune control at the site of virus deposition because saliva or salivary proteins have prevented immune activation. This work is still ongoing, and indeed, research on phleboviruses and sandflies is just beginning to be expanded. There is an expansion in research of the skin stages of arbovirus infection that precede the development of symptoms, which will further inform mechanisms that may be intercepted by salivary proteins.

Remarkably, the sialome among BFA shows remarkable functional similarity between species, despite a relative lack of structural or sequence homology. While it would be ill-advised to assume that all salivary enhancement functions found in one vector are mirrored in another, it may be worth looking more broadly at different mechanisms found amongst hematophagous arthropods to provide insights into viral enhancement shared between arboviral diseases.

## Figures and Tables

**Figure 1 ijms-22-09173-f001:**
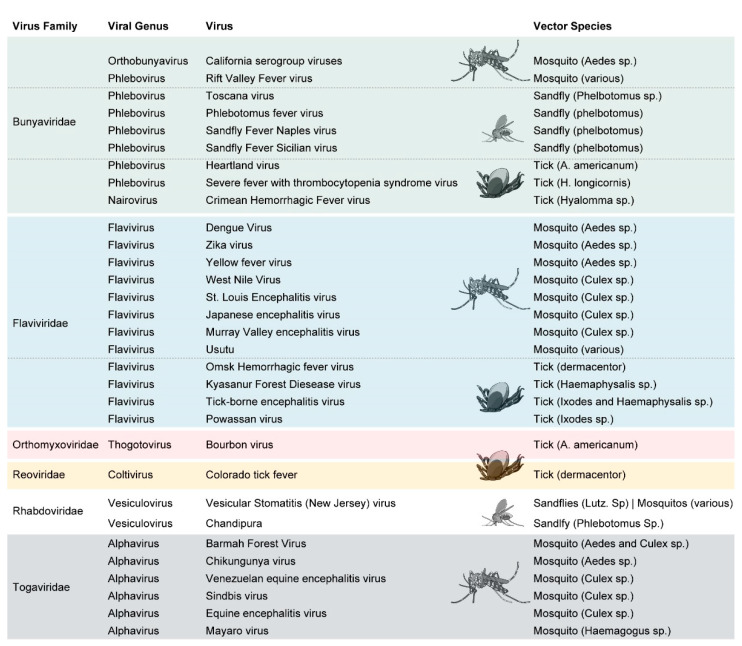
Virus families and common arthropod vectors of viral disease.

**Figure 2 ijms-22-09173-f002:**
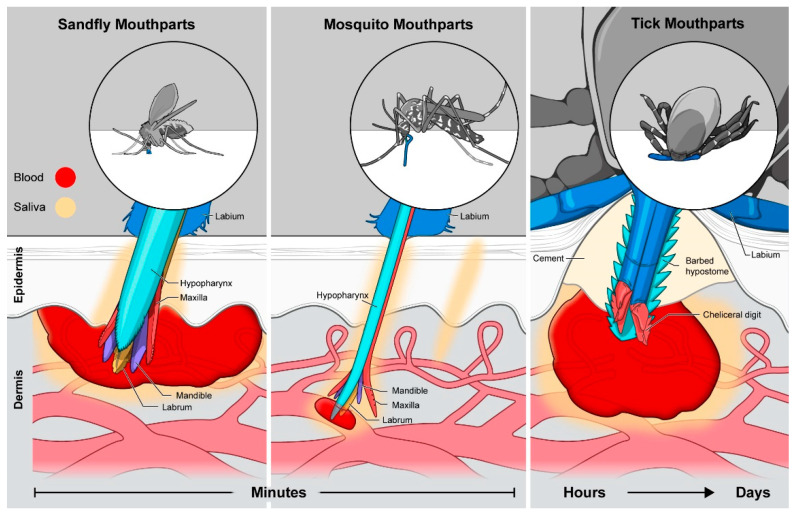
Sandfly, mosquito, and tick feeding methods and mouthparts. Each vector employs a feeding style influenced by the arrangement of their mouthparts. Saliva (shown in yellow) is secreted into the skin by each of the three arthropod vectors.

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
