# Peer review of "Arboviruses: How Saliva Impacts the Journey from Vector to Host"

_ijms, 2021, doi:10.3390/ijms22179173_

Round 1
Reviewer 1 Report
The review entitled ‘Arboviruses: how saliva impacts the journey from vector to host’ provides an overview of arbovirus infection and ways in which arthropod vectors influence viral pathogenesis. The authors focused on how saliva and salivary gland extracts from mosquitoes, sandflies and ticks impact the immune response to arbovirus infection in the skin. The manuscript is well written and presents relevant and recent references. However, the complement system, a crucial component of innate immune response was not mentioned in the work. Complement system is present in blood of the vessels in dermis and plays a key role in host defense against pathogens, including viruses, and viruses are exposed to it during vectors hematophagy. Activation of complement in the skin during the vector bite leads to production of molecules that mediate chemotaxis, opsonization, phagocytosis induction, virolysis by the membrane attack complex, and inflammation. Several works have demonstrated inhibition of the complement system by saliva of ticks, sandflies and recently in mosquitoes. As complement is related to several immune mechanisms, such as cell recruitment and activation, it its important to be mentioned in the text the possible impact of salivary complement inhibitors in arbovirus transmission.
Minor revisions:
Line 120: LN must be limphnode (LN) since it is the first time that it is mentioned in the text.
Line 134: Aedes, Culex and Anopheles are mosquitoes Genera, not Families.
Line 138: “mosquito families” should be mosquito genera or species.
Lines 139: “mosquitos” should be mosquitoes.
Lines 161, 163 and 164: Ixodidae and Argasidae should not be in italic.
Line 210: should be leishmaniasis (name of the disease).
Line 254: dendric cells must be dendritic cells
Line 281: should be “rest of the body”.
Line 319: Leishmania major should be in italic. The species Leishmania major is found in the Eastern Hemisphere, so “worldwide” seems inappropriate. Leishmania sp. would be better.
Line 322 and 386: Leishmania should be in italic.
Line 397: L. major should be in italic.
Line 406: Culex should be in italic.
Author Response
We agree with the reviewer that we were remiss in not including a section on how vector saliva can impact the complement system and how that may affect virus infection. We have corrected this in the revised manuscript.
Minor revisions:
We made all of the minor text changes recommended by the reviewer and appreciate the corrections.
Reviewer 2 Report
Schneider et al. present a scholastic summary of the roles of saliva and salivary proteins in determining the success of feeding and virus transmission by blood feeding arthropods. It is a comprehensive coverage of the literature that I would gladly recommend to vector biology colleagues wishing to dip their toes in the subject. The article makes an enjoyable read and I learned a lot from it.
Lines 120–121. The mechanism the authors refer to in the preceding sentences involves virus replication and dissemination. It's not clear how this mechanism can explain the spread of non-viremic THOGV, assuming non-viremic means non-replicating.
Lines 168–168. Are there sex variations in tick sialome? If yes, are they relevant for virus transmission?
Perhaps section 4.3.1. should be retitled “Innate immune suppression” or “Modulation of innate immune activation” to more accurately reflect the content. Similarly, Section 4.3.2. could instead be “Alterations of cytokine production”.
Line 342–343. “differential cytokine profiles observed during viral exposure in the presence of salivary proteins from different vectors”?
Section 4.3.4. describes the enhancement of host immune response by sandfly SGE. This seems out of place with the rest of section 4.3., which describes immune suppression. The authors could consider reorganising it into Section 4.4. “Chemotaxis to bite site by sandfly SGE”.
On Section 4.3.4.: Are differences between the sialome of phlebovirus- and leishmania-infected sandflies?
Line 385–386. “This was suggested to increase the potential spread of the human parasite Leishmania, as the parasite preferentially replicates within macrophages.” This may be common immunology knowledge but it is not clear to me why attracting monocytes would also attract macrophages.
Line 387. “injection of salivary gland lysates alongside parasites”?
Lines 436–437 “sandflies also suppress cell chemotaxis […] with their saliva” seems contradictory to section 4.3.4. and should be revised here.
Section 4.8 is misnumbered. The title could be revised to be more descritive i.e. “Modulations of adaptive immunity”
Towards the end, I wished there were a few words describing the current and upcoming avenues of research in this topic, i.e., what exciting questions are currently being worked on? What implications would answering these questions have on arbovirus disease control?
There are a few minor editorial fixes to be made that will probably be picked up by the journal’s editors anyway, not limited to but including:
- Line 49. “in cases of severe febrile illness”
- Line 50. “mosquito vector populations”
- Line 53. “sandfly -borne viruses” has an extra space
- Line 120. “LN” is the first instance of this acronym, it should be defined. It is instead defined later and over several times throughout the text.
- Line 288. “co-localizinges”
- Line 291. “CCHFV” is not defined. Crimean-Congo Hemorrhagic Fever virus? In Table 1, this is listed as simply “Crimean Hemorrhagic Fever virus” instead of its full name.
- Line 347–348. “a potent inhibitor blocking of activation of the lymphotoxin B receptor”. Awkward phrasing.
- Inconsistent use of Oxford commas.
- Italicisation of genus and species names to be checked.
Congratulations to the authors for completing this review. It would be a valuable resource for the community.
Author Response
Lines 120–121. The mechanism the authors refer to in the preceding sentences involves virus replication and dissemination. It's not clear how this mechanism can explain the spread of non-viremic THOGV, assuming non-viremic means non-replicating.
Non-viremic does not mean non-replicating. It just means that the virus did replicate in the host to sufficient levels to be detected in the blood stream. Rather the virus is most likely picked up in the skin tissue (localized transmission) by ticks feeding in the same area. We have modified the text to clarify this point. (lines 117-119)
Lines 168–168. Are there sex variations in tick sialome? If yes, are they relevant for virus transmission?
There are a few reports of sex variations in the tick sialome, but no information on whether this could affect virus transmission. We have included this point in the revised manuscript (line 200-201)
Perhaps section 4.3.1. should be retitled “Innate immune suppression” or “Modulation of innate immune activation” to more accurately reflect the content. Similarly, Section 4.3.2. could instead be “Alterations of cytokine production”.
We appreciate the suggestion and have renamed these sections.
Line 342–343. “differential cytokine profiles observed during viral exposure in the presence of salivary proteins from different vectors”?
We have corrected this statement to read more clearly (line 399-400)
Section 4.3.4. describes the enhancement of host immune response by sandfly SGE. This seems out of place with the rest of section 4.3., which describes immune suppression. The authors could consider reorganising it into Section 4.4. “Chemotaxis to bite site by sandfly SGE”.
We appreciate the suggestion and have reorganized this section.
On Section 4.3.4.: Are differences between the sialome of phlebovirus- and leishmania-infected sandflies?
Those studies have not been done. We have included a line on the importance of this type of study (lines 496-498)
Line 385–386. “This was suggested to increase the potential spread of the human parasite Leishmania, as the parasite preferentially replicates within macrophages.” This may be common immunology knowledge but it is not clear to me why attracting monocytes would also attract macrophages.
Monocytes when they are recruited to tissues generally differentiate into macrophages. We have clarified this in the manuscript (line 483)
Line 387. “injection of salivary gland lysates alongside parasites”?
We have clarified this line (line 483)
Lines 436–437 “sandflies also suppress cell chemotaxis […] with their saliva” seems contradictory to section 4.3.4. and should be revised here.
We have clarified this line (line 545)
Section 4.8 is misnumbered. The title could be revised to be more descritive i.e. “Modulations of adaptive immunity”
We have changed the numbering and the title for this section.
Towards the end, I wished there were a few words describing the current and upcoming avenues of research in this topic, i.e., what exciting questions are currently being worked on? What implications would answering these questions have on arbovirus disease control?
We appreciate the suggestion and have added a few lines to the end of the first paragraph of Section 5 (lines 551-555).
There are a few minor editorial fixes to be made that will probably be picked up by the journal’s editors anyway, not limited to but including:
We have made all the minor corrections suggested by the reviewer.
Congratulations to the authors for completing this review. It would be a valuable resource for the community.